# Supramolecular Aggregation of Nanoparticles on Aluminum and Gold Surfaces Occurring in Bovine Serum Albumin Solutions

**Aleksei Salanov \*, Alexandra Serkova, Anastasia Zhirnova, Larisa Perminova and Galina Kovalenko \***

Boreskov Institute of Catalysis, 630090 Novosibirsk, Russia; serkova@catalysis.ru (A.S.);
a.zhirnova@g.nsu.ru (A.Z.); perminova@catalysis.ru (L.P.)
\* Correspondence: salanov@catalysis.ru (A.S.); galina@catalysis.ru (G.K.)

**Abstract:** The supramolecular aggregation processes occurring on metallic (aluminum and gold) surfaces in aqueous solutions of bovine serum albumin (BSA) during drying were studied using advanced scanning electron microscopy (SEM). The possible mechanism for the formation of amazing intricate fractal structures on metallic surfaces was proposed based on the analysis of SEM images, size distribution diagrams and EDX-scanning element distribution maps.

**Keywords:** supramolecular aggregation of nanoparticles; aluminum and gold supports; scanning electron microscopy

## 1. Introduction

Advanced scanning electron microscopy (SEM) opens up new opportunities for visualization of the interaction of various nano-objects, including biological materials (proteins, antibodies, bacteria, etc.), which provides new insights into how biomolecules can form supramolecular 3D structures [1]. The visualization of the aggregation of nanoparticles, with different chemical origins, occurring on different metallic surfaces, is an interesting and informative trend in modern SEM investigations.

The preparation of samples for microscopy requires a special approach and skills, since microdroplets of 1–2 μL in volume are applied to a specially prepared surface of solid substrate (support), followed by their drying. It should be noted that the process of forming thin surface coatings/films with a uniform distribution of the active component, for example, enzymatically active substances, is an important practical problem, in particular, for analytical sensorics. The distribution of such an active component as proteins/enzymes during the drying of their solutions on the substrate surface has been little studied.

Droplets of a solution on a substrate follow one of two drying mechanisms: either the contact line gets pinned and the droplet maintains a fixed contact area (e.g., colloidal dispersions) [2], or the drop maintains a constant contact angle by de-pinning the contact line (e.g., water on non-wetting highly hydrophobic substrates) [3]. During drying of the droplet, the particles of colloidal dispersions deposit in a ring at the periphery of the droplet due to capillary flow, in which the pinned contact line causes the solvent to flow. The authors of [2] studied the formation of thin coatings and films of single-walled carbon nanotubes (SWCNTs) upon drying their aqueous solutions containing F68 Pluronic. Using various microscopy techniques, SWCNTs have been shown to self-assemble into an intricate net-like crust on the surface.

Bovine serum albumin (BSA) is a well-studied protein [4–7]. Protein molecules (M.m. 64–69 kiloDa) are a globule in the form of an oblate ellipsoid of revolution, with crystallographic dimensions (8–13) nm × (4–5) nm (from the DOI database: 10.2210/pdb4F5S/pdb). The BSA molecule has an equal number of hydrophobic and hydrophilic amino acid

residues; isoelectric point (pJ) is 4.8–5.6, i.e., the BSA molecule is neutral in distilled water with a pH~6. The ability of BSA molecules to interact with numerous biologically active substances, such as various metabolites (bilirubin, urobilin, fatty acids, bile salts) and some exogenous substances (penicillin, sulfonamides, mercury) makes it a peculiar "taxi" in the bloodstream [4,6]. The ability of BSA molecules to interact with numerous biologically active substances makes it possible to develop efficient and biospecific adsorbents based on immobilized serum albumin [8–13]. The adsorption of BSA molecules was investigated by scanning electron microscopy (SEM), Fourier transform infrared (FTIR) spectroscopy and X-ray photoelectron spectroscopy (XPS) [12]. BSA molecules are capable of aggregation and these processes are well studied [14,15]. At a neutral pH, hydrophobic–hydrophobic interactions predominate between the globules, leading to the formation of disordered aggregates. At a high protein concentration, partially folded conformers form oligomers (dimers, tetramers, etc.), which, at elevated temperatures, turn into ordered amyloid-like fibrils [15,16]. By manipulating the amyloid-like aggregation of proteins, in particular, bovine serum albumin, the authors of [16] developed a highly efficient biosorbent to extract precious metals, such as Au, Ag, Pd, Pt and Ir, from ore and electronic equipment. Due to the rapid aggregation of BSA via a reduction in intramolecular disulfide bond, a sandwich-like membrane was obtained, capable of adsorbing gold in the amount of 316 mg/m$^2$, which is ~2100-times higher than that characteristic of widely used industrial adsorbents, such as activated coal and resin [16].

In this paper, a study was launched to investigate the supramolecular aggregation of nanoparticles from aqueous solutions of BSA on two metallic surfaces. An aluminum disc and a gold film were used as the substrates (supports) for microdroplets of protein solutions, which were then dried. Supramolecular aggregates formed on metallic surfaces were visualized by scanning electron microscopy, equipped with EDX analysis apparatus. The phenomena of nanoparticle aggregation and BSA distribution over a dried microdroplet were described based on the analysis of SEM images, particle size distribution diagrams, and EDX analysis data.

## 2. Materials and Methods

An ultra-high-resolution field emission scanning electron microscope (SEM) Regulus 8230 (Hitachi, Tokyo, Japan) was used for the investigations at $E_0$ = 5 keV in secondary electrons (SE) mode. An aluminum disk was polished using standard equipment for material polishing MultiPrep (Allied High-Tech Products, Inc., Rancho Dominguez, CA, USA). A gold film with a thickness of ca. 10 nm was supported on a quartz plate by magnetron sputtering of Au using a JFC-1600 Auto Fine Coater (Jeol, Tokyo, Japan).

Bovine serum albumin produced by SIGMA Co. was dissolved in distilled water. Solutions with different protein concentrations from 0.1 to 2.0 mg mL$^{-1}$ were used for sample preparation for microscopy: a 2 µL droplet of BSA solution was applied to the metallic (Al or Au) substrate and then dried for several hours in a desiccator under ambient conditions (20 ± 2 °C, 1 bar).

## 3. Results

*SEM Study*

SEM study was carried out to elucidate the features of the aggregation processes of nanoparticles inside microdroplets of BSA solution dried on metal surfaces in a desiccator under ambient conditions. Typical SEM images of impressive intricate fractal structures formed on Al and Au surfaces are shown in Figures 1 and 2.

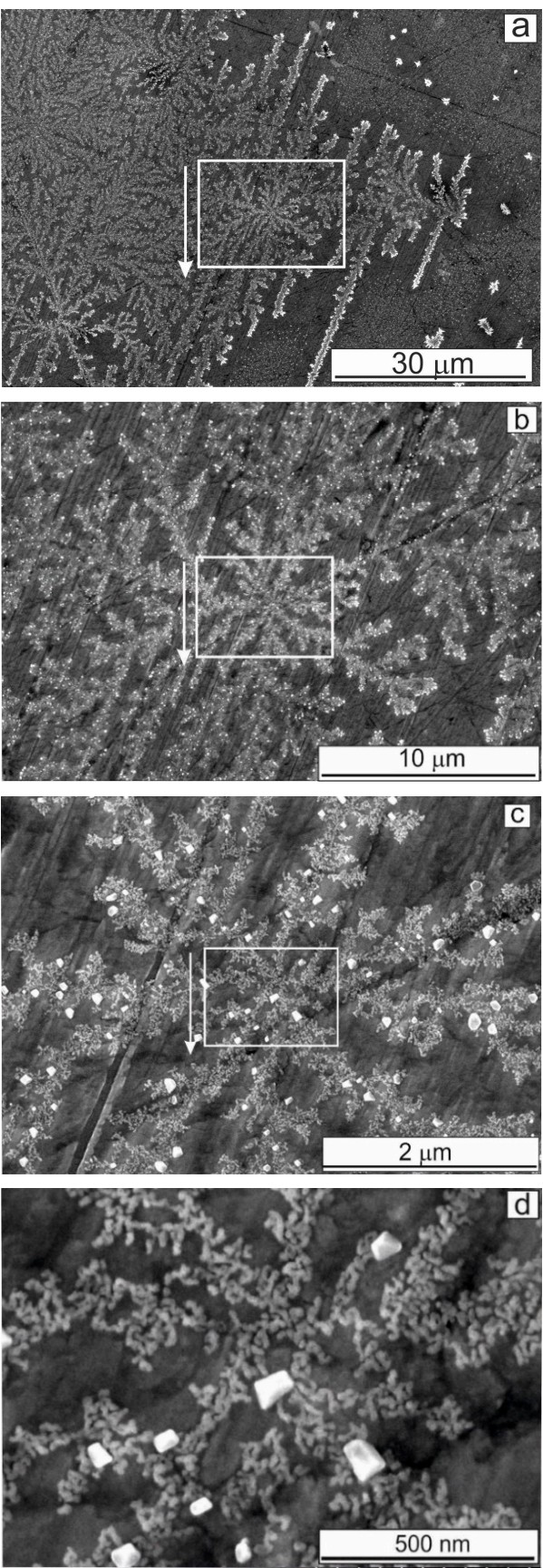

**Figure 1.** (**a–d**)–SEM images of supramolecular nanoparticle aggregates formed on the Al disk with different magnification, obtained in the SE mode. Rectangles and arrows indicate areas with a higher magnification.

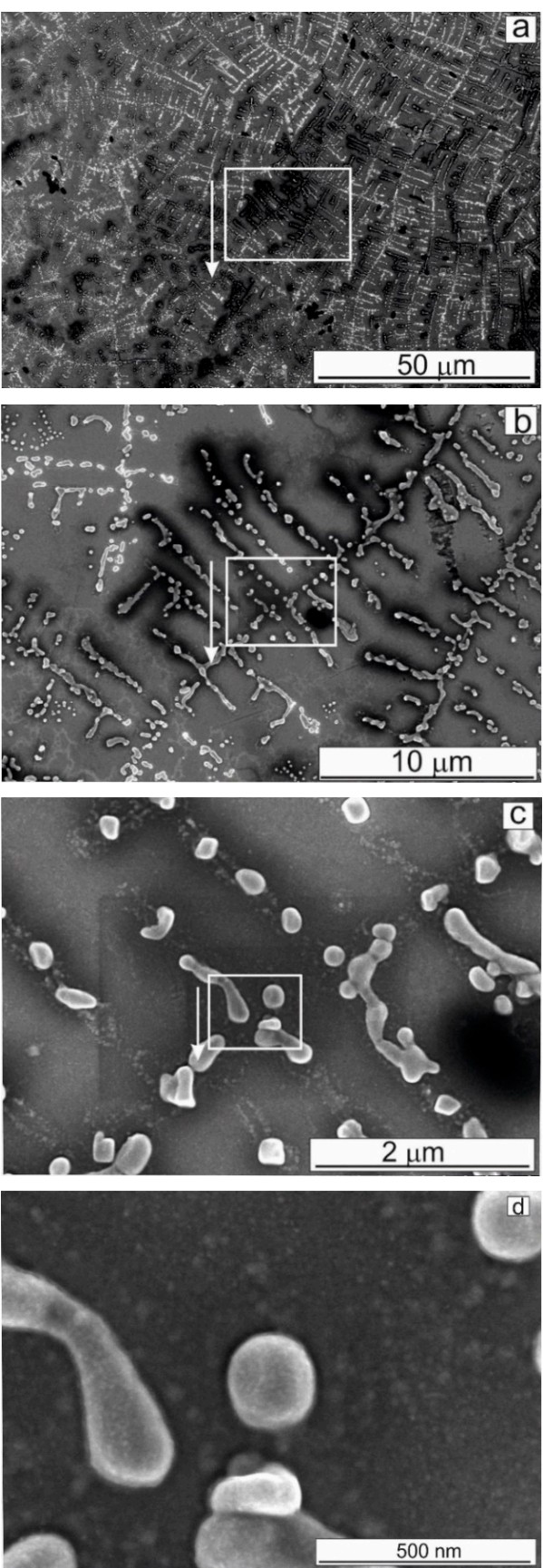

**Figure 2.** (**a–d**)–SEM images of supramolecular nanoparticle aggregates formed on the Au film with different magnification, obtained in the SE mode. Rectangles and arrows indicate areas with a higher magnification.

For aluminum, the SEM images were first processed to obtain diagrams of the size distribution of single nanoparticles (Figure 3). Two peaks in the size of nanoparticles were observed: one of them corresponded to the small nanoparticle, 10–12 nm in size (Figure 3a), while another, to 20–30 nm (Figure 3b). The primary single nanoparticles formed flower-like objects, 0.5–3 μm in size, which were located near the fractals (Figure 1a); the surrounding area of these "flowers" was empty. The elongated branches of the fractals were formed and located along the visible extended surface defects (Figure 1b,c). Cubic-shaped nano-objects ca. 100 nm in size were formed over the branches at the ends (Figure 1d). Surprisingly, the chemical composition of these bright objects was consistent with sodium chloride (NaCl), as EDX analysis showed. The possible mechanism of supramolecular aggregation on the Al surface was proposed. During the drying of microdroplets of aqueous BSA solutions, the 10–12 nm primary single nanoparticles formed, which then organized into dimers and bigger oligomers, and further into flower-like nucleus objects, 1–3 μm in size. Fractal structures were formed along surface defects, which undoubtedly played an important role in the processes of supramolecular aggregation on the surfaces.

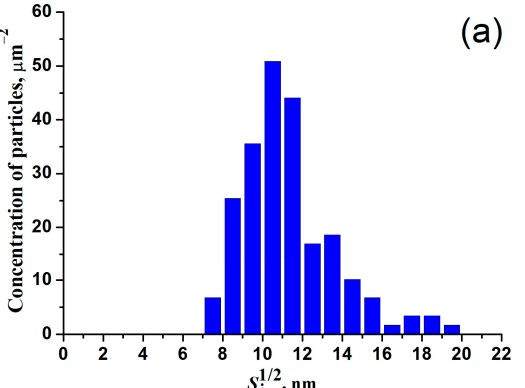 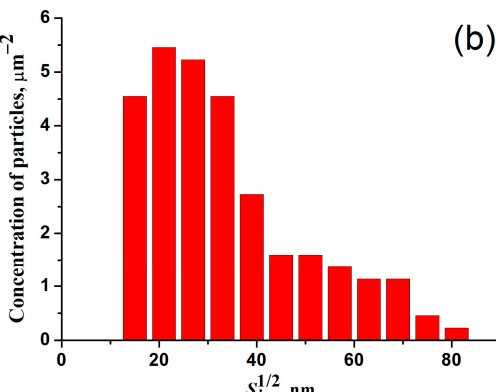

**Figure 3.** The diagram's size distribution of the single nanoparticles formed on aluminum: (**a**)—size distribution of smallest single nanoparticles on the Al surface with an area of 0.59 μm$^2$ containing 133 nanoobjects; (**b**)—size distribution of biggest single nanoparticles on the surface area with an area of 4.4 μm$^2$ containing 132 objects.

For the gold, peculiar structures with the appearance of "big city streets" were observed (Figure 2), which were not observed on the aluminum surface. The observed supramolecular aggregates consisted of rounded particles with the predominant size above 200 nm (Figure 2c,d). Impressive fractal structures similar to those described above (Figure 1b) also formed on the gold surface.

EDX analysis was applied to determine the chemical composition of both albumin (in powder) and nanoparticle aggregates. For BSA, the following results were obtained (in wt.%): carbon—60.2; oxygen –19.9; nitrogen—17.5; sulphur—1.9; sodium—0.5; chlorine—0.01. As noted above, the chemical composition of the cubic-shaped objects at the ends of the fractal branches (Figure 1d) corresponded to sodium chloride. Where was the albumin localized? Note, that the presence of carbon (C) and nitrogen (N) in the element distribution maps was considered characteristic of the protein molecules. EDX scanning showed that almost all of the protein was at the edge of the droplets (Figure 4), and the greater the protein concentration in the solution, the greater the carbon and nitrogen content in the concentric ring. This means that when the droplet dried, the molecules of dissolved albumin concentrated at its edge, since the affinity of BSA for the aluminum surface was low, and capillary forces pulled protein molecules to the edge of the droplet, as described in [2]. Similar element distribution maps were observed for the droplets of BSA solutions on aluminum and gold substrates.

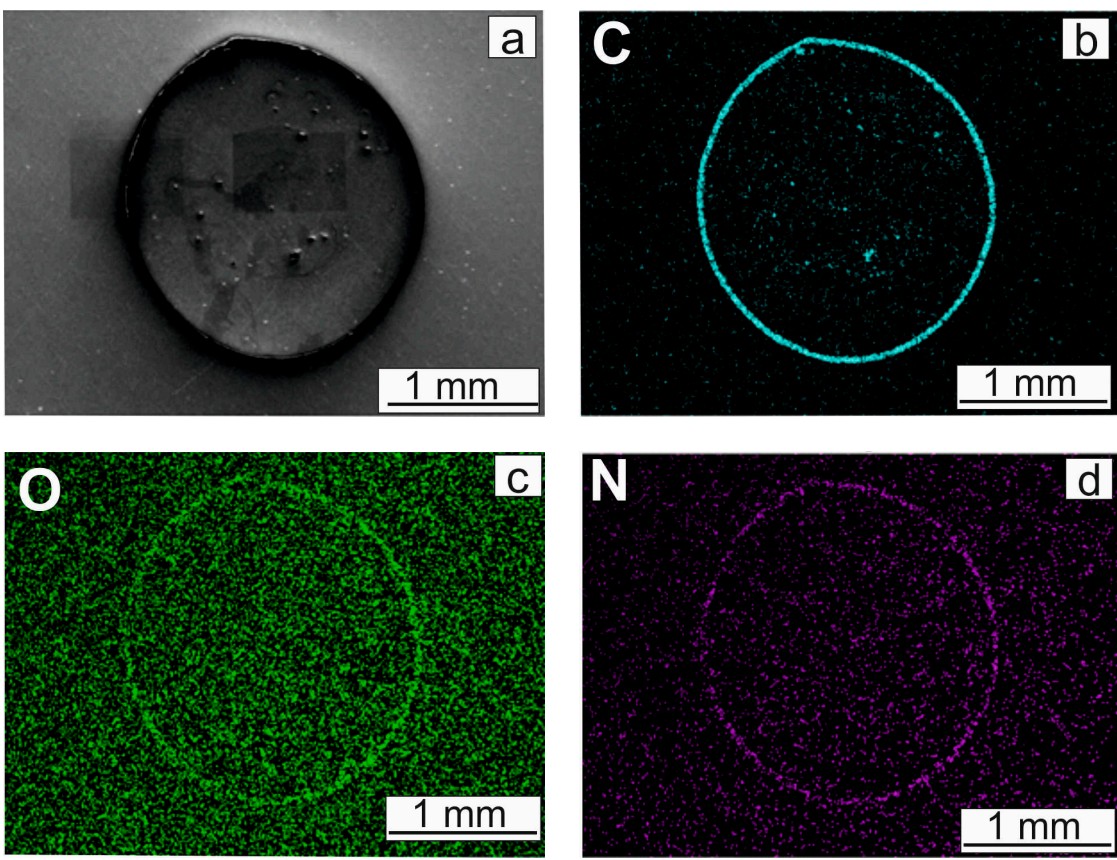

**Figure 4.** (**a**)—SEM image of the microdroplet of BSA solution (0.2 mg·mL$^{-1}$) on aluminum. Element distribution maps corresponding to image (**a**): (**b**)—carbon, (**c**)—oxygen, (**d**)—nitrogen.

Microdroplets from the solution of sodium chloride (0.004 mg mL$^{-1}$ of NaCl without albumin) were dried on the aluminum substrate under similar conditions as above. Interestingly, amazing intricate fractal structures were also observed (Figure 5), but the cubic NaCl crystals shown in Figure 1c,d, were not found. Apparently, the processes of aggregation of nanoparticles during drying were affected by the presence of albumin in the aqueous solution.

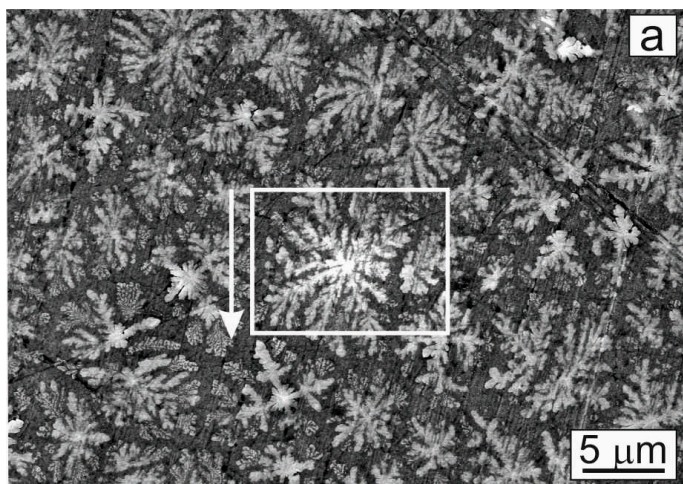

**Figure 5.** *Cont.*

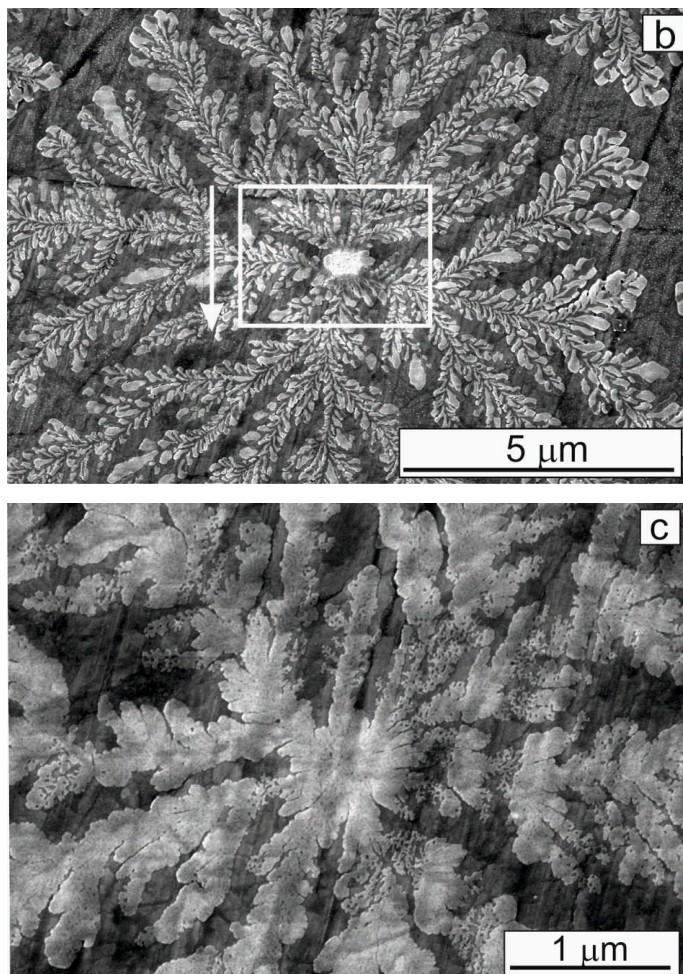

**Figure 5.** (**a–c**)–SEM images of aggregates of nanoparticles formed on an aluminum substrate during drying of microdroplets of NaCl solutions (0.004 mg·mL$^{-1}$ without albumin) at various magnifications, obtained in the SE mode.

## 4. Discussion

The features of the aggregation processes of nanoparticles on Al and Au surfaces were revealed: (i) sizes of the single particles involved in the aggregation processes were 10–30 nm on the Al surface and above 200 nm on the Au surface (Figures 1d and 2d), (ii) the predominant shape of the single nanoparticles differed depending on metallic surfaces, viz., cubic-shaped objects formed over and at the end of the fractal branches on the aluminum, and elliptical or rounded nano-objects on the gold, (iii) the impressive and amazing intricate fractal structures were formed on both surfaces (Figures 1 and 2). The chemical composition of the bright cubic crystals at the ends of the fractal branches that formed on the Al surface corresponded to sodium chloride. The molecules of dissolved albumin concentrated at the edge of the dried droplet due to capillary forces that pulled BSA to the edge.

At present, the study of supramolecular aggregation of nanoparticles using advanced scanning electron microscopy, in particular, on carbon surfaces, such as highly oriented pyrolytic graphite (HOPG), is being carried out.

**Author Contributions:** SEM study and data processing, A.S. (Aleksei Salanov); working on Regulus electron microscope, A.S. (Alexandra Serkova) and A.Z.; preparing the samples (microdroplets) for SEM study, L.P.; writing and supervision, G.K. All authors have read and agreed to the published version of the manuscript.

**Funding:** This work was supported by the Ministry of Science and Higher Education of the Russian Federation within the governmental order for Boreskov Institute of Catalysis (project AAAA-A21-121011390007-7). The studies were carried out using facilities of the shared research center «National Center of investigation of catalysts» at the Boreskov Institute of Catalysis.

**Institutional Review Board Statement:** Not applicable.

**Informed Consent Statement:** Not applicable.

**Data Availability Statement:** Not applicable.

**Conflicts of Interest:** The authors declare no conflict of interest.

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
