# Peer review of "Supramolecular Aggregation of Nanoparticles on Aluminum and Gold Surfaces Occurring in Bovine Serum Albumin Solutions"

_2673-8023, doi:10.3390/micro2020022_

Round 1
Reviewer 1 Report
The work by Salanov et al. focuses on the formation of intricate fractal structures on metallic surfaces by bovine serum albumin. The manuscript is well-written and the presented data is clear and of high quality. While the manuscript could be accepted in its current form, I have a few minor points and suggestions for the authors.
- The introduction states “…aggregation of nanoparticles from aqueous solutions of BSA on various metallic surfaces…” (lines 58-59). However, the manuscript only displays the aggregation on two different metal surfaces (aluminum and gold). I believe the phrasing “various metallic surfaces” may be an overstatement in this case.
- The authors mention that the metallic surfaces have defects, which may play a role in the aggregate formation. Is it possible that there are fractures on the surface, which directly relate to the observed aggregate structures i.e., fractal fractures on aluminum and linear fractures on gold?
- Have the authors considered additional studies on the formed BSA aggregates, which would allude to their amyloid-like nature? As an example, are the formed aggregates capable of an amyloid-like self-replication, when not in the presence of metallic surfaces? Are they capable of binding amyloid-specific dyes (Congo red or thioflavin)?
- The SEM images of BSA show quite a large number of NaCl crystals, which formed during the drying procedure, but the method section indicates that BSA was dissolved in distilled water. The obvious question here is, where did the NaCl come from; was it present in the BSA stock?
- Lastly, there is a lack of information regarding the replicability of the observed structure formation. Do multiple, independent repeats always result in fractal structures on aluminum and linear structures on gold or is the final structure a random occurrence?
Author Response
The Authors sincerely thank the Reviewer for the questions and comments.
The Аuthors’ answers are in file attached

Reviewer 2 Report
The research article “Supramolecular aggregation of nanoparticles on aluminum and gold surfaces occurring in bovine serum albumin solutions,” by Aleksei Salanov et al aims to evaluate the supramolecular aggregation processes occurring on metallic surfaces in aqueous solutions of proteins namely serum albumin. It is a timely article considering the importance of analytical sensorics and it has been little studied. Authors have used advanced techniques like scanning electron microscopy, size distribution diagrams and EDX-scanning elements distribution maps. I am enthusiastic about this article and supportive of its publication. I only offer some minor suggestions to improve readability and enhance the message of the paper (adopting them is optional).
Minor issues:
- Figure 1, 2 and 5. Individual SEM images could be combined in the form of a Panel.
- Figure 3: Spelling of “concentration” wrong in Figure. Also the Axis label should be positioned vertically along the axis. The n , mead and SD should be mentioned.
- Conclusion and future implications of the work missing at the end of the discussion section.
- Authors should consider adding a few more references.
- DLS could be performed.
Author Response
The authors are very grateful to the Reviewer. The answers are in file attached.
